

# Restriction digest screening facilitates efficient detection of site-directed mutations introduced by CRISPR in *C. albicans UME6*

Ben A. Evans[1], Olivia L. Smith[1], Ethan S. Pickerill[1], Mary K. York[1], Kristen J.P. Buenconsejo[2], Antonio E. Chambers[1] and Douglas A. Bernstein[1]

[1] Department of Biology, Ball State University, Muncie, IN, United States of America
[2] Department of Microbiology and Immunology, Drexel University, Philadelphia, PA, United States of America

Corresponding author
Douglas A. Bernstein,
dabernstein@bsu.edu

## ABSTRACT

Introduction of point mutations to a gene of interest is a powerful tool when determining protein function. CRISPR-mediated genome editing allows for more efficient transfer of a desired mutation into a wide range of model organisms. Traditionally, PCR amplification and DNA sequencing is used to determine if isolates contain the intended mutation. However, mutation efficiency is highly variable, potentially making sequencing costly and time consuming. To more efficiently screen for correct transformants, we have identified restriction enzymes sites that encode for two identical amino acids or one or two stop codons. We used CRISPR to introduce these restriction sites directly upstream of the *Candida albicans UME6* $Zn^{2+}$-binding domain, a known regulator of *C. albicans* filamentation. While repair templates coding for different restriction sites were not equally successful at introducing mutations, restriction digest screening enabled us to rapidly identify isolates with the intended mutation in a cost-efficient manner. In addition, mutated isolates have clear defects in filamentation and virulence compared to wild type *C. albicans*. Our data suggest restriction digestion screening efficiently identifies point mutations introduced by CRISPR and streamlines the process of identifying residues important for a phenotype of interest.

## BACKGROUND

Site-directed mutagenesis may be used to introduce a mutation that leads to a change in protein amino acid sequence (*Shortle, DiMaio & Nathans, 1981*). Investigators can then assess the role of the altered residue/s in particular phenotypes. Site-directed mutagenesis has been used to identify and characterize enzyme active sites, assess the role of amino acid modification on protein function, and characterize a wide variety of additional protein properties (*Winter et al., 1982*).

Alanine-scanning mutagenesis, where a particular amino acid/s is changed to alanine, is a particularly popular application of this technique (*Cunningham & Wells, 1989*; *Weiss et al., 2000*). The methyl sidechain of Alanine is small and nonpolar. The benign nature

of the alanine sidechain often abolishes the functionality of the original residue while minimally disrupting protein backbone structure. Additional protein variants can be made in analogous fashions. For instance, in many cases, the carboxyl sidechain of a glutamate residue biochemically mimics a phosphate. Thus, mutation of a threonine or serine to glutamate mimics constitutive phosphorylation (*Szewczuk, Tarrant & Cole, 2009*; *Thorsness & Koshland, 1987*). If an investigator hypothesizes an aromatic or positively charged residue interacts with nucleic acid, a negatively charged or small hydrophobic residue can be introduced to test if said residue is important for nucleic acid binding (*Bernstein & Keck, 2005*). Moreover, stop codons can be introduced to an open reading frame to generate a C-terminal truncation and a stop codon introduced at the beginning of the open reading frame will terminate translation before a functional protein is made, generating a null protein phenotype. Furthermore, transcription initiation (*Vo Ngoc et al., 2017*) and splicing (*Bortfeldt et al., 2008*) are controlled in part by short nucleotide sequences that can be readily interrogated by site-directed mutagenesis. As such, site-directed mutagenesis is a powerful tool to elucidate not only protein but DNA and RNA sequence function.

Historically, site-directed mutagenesis has been a multistep process. In yeast for instance, cells are transformed with exogenous DNA containing a mutation and homologous recombination incorporates the mutant sequence into the genome. Alternatively mutations can be subcloned onto plasmids and transformed into yeast lacking the gene of interest (*Cormack & Castano, 2002*). These techniques require cells perform homologous recombination at a relatively high rate or maintain plasmids. In addition, detection of single nucleotide polymorphisms (SNPs) between alleles in a diploid is challenging. Incorporation of restriction enzyme cut sites via PCR has been used to detect SNPs using derived cleaved amplified polymorphic sequence analysis (*Hodgens, Nimchuk & Kieber, 2017*; *Neff et al., 1998*).

One organism that has proven challenging to modify by site-directed mutagenesis is the human fungal pathogen *Candida albicans*. Because *C. albicans* is diploid, introduction of a homozygous point mutant has historically required multiple rounds of transformation and homologous recombination (*Jones et al., 2004*; *Muzzey et al., 2013*). Furthermore, plasmids are not maintained consistently in *C. albicans*. The development of the *Candida* CRISPR system however has made introduction of mutations into the genome more efficient (*Vyas, Barrasa & Fink, 2015*; *Vyas et al., 2018*). During CRISPR-mediated genome editing, Cas9 nuclease bound to a guide RNA molecule targets a specific genomic sequence to be modified using base pairing. Once bound to the complementary sequence, Cas9 introduces a double-strand break. To introduce the desired mutation and repair the double-strand break by homologous recombination, a double-stranded DNA repair template possessing the altered sequence is cotransformed (reviewed in *Dominguez, Lim & Qi, 2016*; *Wang, La Russa & Qi, 2016*). Clones are sequenced to determine if the desired mutation has been introduced, but efficiency of repair template incorporation can be a substantial bottleneck. Furthermore, DNA sequencing is required to ensure the correct mutation has been introduced into the genome.

We have developed a system to inexpensively screen potential point mutants using restriction digestion. The system enables the investigator to screen for the introduction of two alanine, two glutamate, two arginine, two glycine or one or two stop codons using colony PCR and restriction digestion. As proof of concept, we introduced mutations into *UME6*, a transcription factor that regulates *C. albicans* filamentation and virulence (*Banerjee et al., 2008*) and screened for repair template incorporation by PCR and restriction digestion. Specifically, we introduced mutations upstream of sequence encoding the Ume6 $Zn^{2+}$ finger DNA-binding domain. We found the frequency different repair templates were incorporated into the genome varied significantly. We characterized the effects these mutations had on *C. albicans* filamentation. Strikingly, introduction of a stop codon directly upstream of the *UME6* $Zn^{2+}$ finger DNA binding domain eliminated filamentation. In addition, we found mutation of two conserved positively charged residues to glutamate directly upstream of the *UME6* $Zn^{2+}$ finger DNA-binding domain resulted in filamentation defects and a decrease in *C. albicans* virulence.

## MATERIALS AND METHODS

### Cloning and sequencing

Primers used in this study are listed in (Table 1). Ume6 guide RNA primers (Ume6 Guide 2 fr and rv) were subcloned into pv1093 as previously described (*Vyas, Barrasa & Fink, 2015*). Repair templates were made by PCR using indicated primer pairs. PCR products were purified using a Zymo PCR Purification Kit. Lithium Acetate transformation was performed and cells were plated on yeast extract-peptone-dextrose agar plates that contained 200 μg/ml nourseothricin (Nat) (*Gietz & Woods, 2002*). All growth media for this study was supplemented with 27 mM uridine. After three days, Nat$^r$ colonies were streaked for isolation on YPD medium supplemented with 100 μg/ml Nat. Colony PCR was performed using primers Ume6 ch primer 2 fr and rv. 10 μl of colony PCR product was digested in accordance with the manufacturer's recommended reaction conditions. Digestion and undigested PCR products were resolved on a 2% agarose gel, stained with ethidium bromide, and visualized. PCR products with digestion patterns consistent with successful mutagenesis were sequenced by GeneWiz, and SNAPgene was used to analyze the sequences.

### Growth and filamentation assays

*C. albicans* strains were grown overnight at 25 °C in YPD and diluted to OD 0.1. Fourfold serial dilutions were plated using a pin replicator on YPD and Spider (*Liu, Kohler & Fink, 1994*). Liquid filamentation was assessed by growing *C. albicans* variants for 24 h in Spider Media at 37 °C.

### *Galleria mellonella* Infection

Healthy (235 mg ± 45 mg) larvae were randomly assigned to sterile PBS, wild type, or mutant groups. Log-phase cultures of *C. albicans* grown at 37 °C in YPD were washed twice and suspended in sterile PBS. Prior to each injection, a Hamilton syringe was sterilized via rinsing once with ethanol and then twice with sterile PBS. Larvae were swabbed with 70%

**Table 1 Oligos used in this study.** Restriction Sites Sequences are bolded and capitalized in repair templates. Nucleotides added for cloning into the CRISPR plasmids are bolded and italicized in the guide primer sequences.

| Oligo name | Primer sequence | Restriction site |
|---|---|---|
| Ume6 Guide 2 fr | ***ATTTG***tacttctacttctaatccaa***G*** | XXXXXX |
| Ume6 Guide 2 rv | ***AAAAC***ttggattagaagtagaagta***C*** | XXXXXX |
| Ume6 ch primer 2 fr | ggtcatgatcatgatgatgaaaat | XXXXXX |
| Ume6 ch primer 2 rv | ctccacaaattggtgtgacttc | XXXXXX |
| Ume6 rp 2_Glu fr | atggcactaacaccaatactgattctacttctacttctaatccaatggtg**GAGGAG**aaac | BseRI |
| Ume6 rp 2_Glu rv | catccttttttagatctaggtaataatcttcttcttgtatgttt**CTCCTC**caccattgga | BseRI |
| Ume6 rp 2_TAA fr | aatggcactaacaccaatactgattctacttctacttctaatccaatggt**TTAATTAA**aa | Pac1 |
| Ume2 rp 2_TAA rv | tccttttttagatctaggtaataatcttcttcttgtatgttt**TTAATTAA**accattggat | Pac1 |
| Ume6 rp 2_TGA fr | atggcactaacaccaatactgattctacttctacttctaatccaatggt**CTGAAG**aaaca | Acu1 |
| Ume6 rp 2_TGA rv | acatccttttttagatctaggtaataatcttcttcttgtatgttt**CTTCAG**accattgga | Acu1 |

ethanol. 10 μl of either sterile PBS or $10^5$ cfu of *C. albicans* suspended in sterile PBS was injected per larva into the left rear proleg. Larvae were incubated at 37 °C in a glass petri dish, and survival was assessed daily by visual inspection and prodding with a sterile pipet tip. Deceased larvae were removed from the dish daily.

## RESULTS

One of the most challenging and costly aspects of site-directed mutagenesis is screening transformants to identify those harboring the intended mutation. To reduce cost and increase the speed by which transformants can be screened, we identified restriction enzyme recognition sequences that, when introduced, convert two codons to codons encoding two alanines, two arginines, two glutamates, two glycines, stop codons (Table 2). To test our strategy, we cotransformed wild type *C.albicans* with a plasmid encoding Cas9 and a *UME6*-targeting guide RNA and PCR-generated repair template DNA with *BseRI*, *PacI*, or *AcuI* recognition sequences. Template-mediated repair replaced *UME6* residues 2259-2265 (encoding Lys754 and Lys755) with either two negatively charged glutamate residues (repairEE; *BseRI* sequence), a TAA stop codon (repairTAA; *PacI* sequence), or a TGA stop codon (repairTGA; *AcuI* sequence) (Fig. 1, Table 2).

Each transformation resulted in dozens of colonies. We used colony PCR to amplify *UME6* from each transformant and digested the PCR products with the enzyme predicted to cut the repair template. 2 of 48 transformants transformed with the repairEE template (designed to replace Lys754/755 with glutamates) were successfully digested by BseRI. 2 of 8 transformants repaired with the repairTAA template (designed to introduce the TAA stop codon) were successfully digested with PacI, and 2 of 48 transformants repaired with repairTGA (designed to introduce the TGA stop codon) were successfully digested with AcuI (Fig. 1B). We sequenced transformants with restriction digestion patterns consistent with successful mutagenesis. We found successfully digested clones had correctly introduced homozygous point mutations in the *UME6* gene (Figs. 1D and 1E). In addition, we did not

**Table 2  Restriction sites that encode consecutive amino acids or stop codons.** Nucleotide code is as follows: **W** = A or T, **M** = A or C, **K** = G or T, **R** = A or G, **N** = A T C or G, / = cleavage site. Numbers in parentheses correspond to how far downstream a cut site will occur from the recognition sequence.

| Translation product | Codon | Restriction enzyme recognition site sequence |
|---|---|---|
| 2×Alanine | GCN | **BbvI** GCAGCN (8/12), **Fnu4HI** GC/NGCN, **TseI** G/CWGCN |
| 2×Arginine | CGN | **Hpy99I** CGWCGN/ |
| 2×Glutamate | GAR | **BseRI** GAGGAG (10/8) |
| 2×Glycine | GGC+GGA | **EciI** GGCGGA (11/9) |
| 1×Stop Codon | TAG | **SpeI** A/CTAGT |
| 1×Stop Codon | TAA | **PacI** TTAAT/TAA |
| 1×Stop Codon | TGA | **AcuI** CTGAAG(16/14) |
| 2×Stop Codon | TAG+TAA | **I-SceI** TAGGGATAACAGGGTAAT(-9/-13) |

observe unintended mutations in *UME6,* and we did not recover heterozygous mutants. We also sequenced PCR products that did not digest; as predicted, those transformants retained wild type *UME6* sequence (Fig. 1C). Names and genotypes of strains used in further studies are listed (Table 3). Our data establish that restriction digestion is a rapid and accurate screening strategy for site-directed mutagenesis in *C. albicans.*

Ume6 is a transcription factor important for *C. albicans* filamentation and the only domain within Ume6 predicted by sequence homology is a $Zn^{2+}$ finger DNA-binding domain (amino acids 760–810) (*Banerjee et al., 2008*). Ume6's C-terminal region (amino acids 811–844) is highly conserved, but is not predicted to form a known structural domain (*Inglis et al., 2012*). We hypothesized the $Zn^{2+}$ finger and C-terminus of Ume6 are important for filamentation. We found neither yeast harboring the premature stop codon (*ume6-stopTAA)* nor those expressing the lysine-to-glutamate point mutants (*ume6-EE*) exhibited slow growth relative to wild type *C. albicans* at a range of temperatures on rich media (Fig. 2B). However, we did find wild type yeast display a more pronounced crinkled colony morphology than either *ume6-EE* or *ume6-stopTAA* yeast on rich media at 37 °C (Fig. 2B).

Wild type, but not homozygous *ume6/ume6 C. albicans* undergo filamentation when cultured on spider medium (*Banerjee et al., 2008*). We tested if the DNA-binding domain and conserved upstream positively charged residues Lys754 and Lys755 are important for induction of *C. albicans* filamentation. We analyzed filamentation of wild type, *ume6-EE*, and *ume6-stopTAA* yeast on spider agar media. We found that, while wild type yeast filament robustly after 7 days, *ume6-stopTAA* yeast failed to filament (Fig. 2A). Furthermore, we found *ume6-EE* yeast exhibit intermediate filamentation (Fig. 2A). We observed similar filamentation patterns when these strains were cultured in liquid spider media (Fig. 2C).

Next, we tested the effect of our point mutations on *C. albicans* virulence in a wax moth model. *ume6/ume6* yeast exhibit lower virulence in a mouse model (*Banerjee et al., 2008*), and their virulence has not been tested in the wax moth model system. Less than 30% of larvae injected with wild type *C. albicans* survived after incubation for 200 h. By comparison, the *ume6-stopTAA* mutation significantly decreases virulence, with roughly

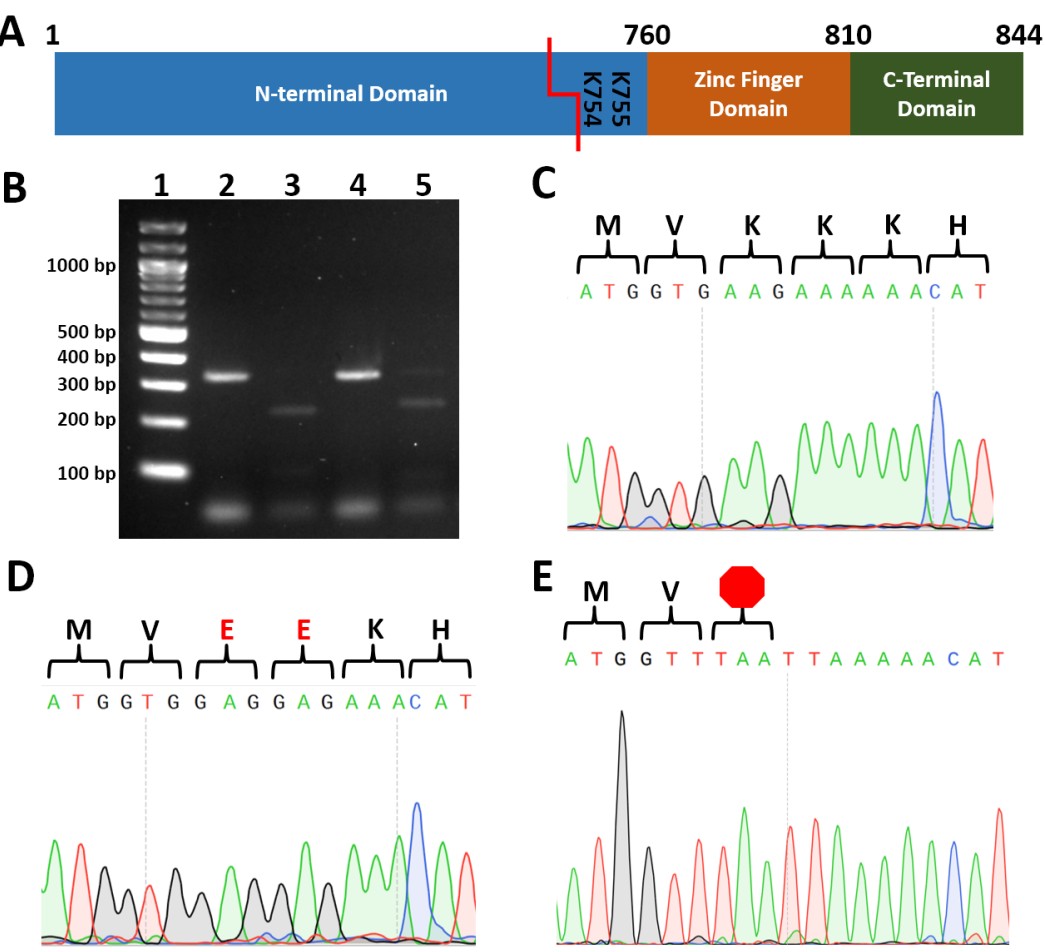

**Figure 1** **Introduction of restriction sites enable efficient screening for correct point mutants.** (A) Cartoon of Ume6 domains and key residues for this study. Red line indicates CRISPR cut site. (B) Restriction digestion of colony PCR of *UME6* from representative *ume6TAAStop* and *ume6-EE C. albicans*. Lane **1.** Ladder, **2.** *ume6TAAStop* undigested, **3.** *ume6TAAStop* digested with Pac1 **4.** *ume6-EE* undigested **5.** *ume6-EE* digested with BseRI **C-E.** Sequence analysis of site of *ume6* mutagenesis of representative failed mutant (wild type sequence) (C), *ume6-EE* (D), and *ume6TAAStop* (E) *C. albicans* amino acid sequence is presented above DNA sequence. Red E designates site of lysine-to-glutamate point mutations. Red octagon designates site of introduced stop codon.

**Table 3** **Strains used in this study.**

| Name | Genotype | How strain is referenced in text |
|------|----------|----------------------------------|
| SC5314 | *UME6:UME6* | Wild Type |
| DAB898 | *ume6-LYS754E LYS755E:ume6-LYS754E LYS755E* | *ume6-EE* |
| DAB894 | *ume6-LYS754stop:ume6-LYS754stop* | *ume6-stopTAA* |

80% surviving after 200 h of incubation (Fig. 2D). Consistent with our filamentation data, we found *ume6-EE* yeast exhibited virulence intermediate to wild type and *UME6stop-TAA* yeast, with greater than 40% of larvae surviving following 200 h of incubation. Our data indicate the final 100 amino acids of Ume6 that include both a $Zn^{2+}$ finger domain and

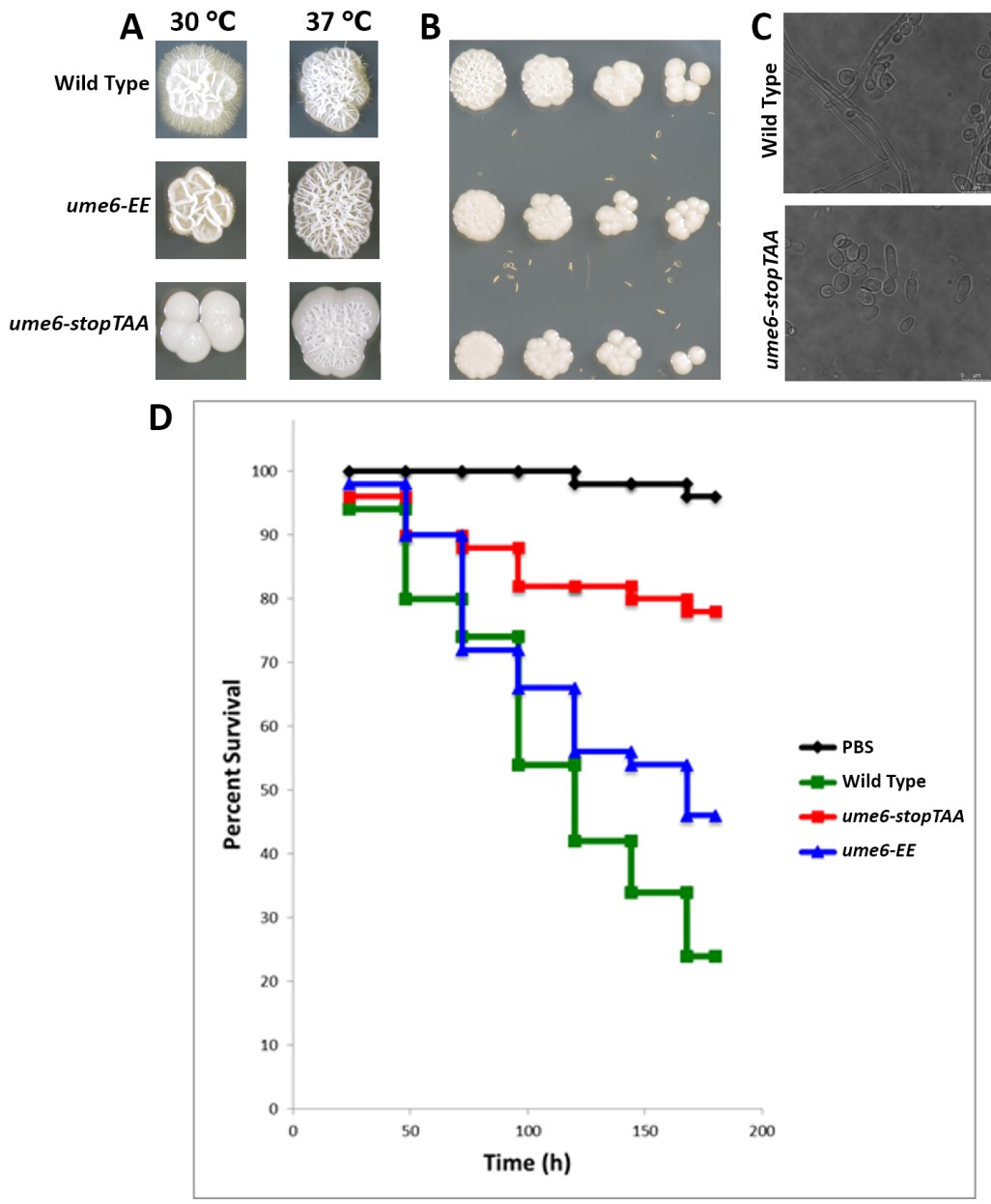

**Figure 2  Assessment of UME6 mutations on filamentation and virulence.** (A) Wild type, *ume6-EE*, and *ume6-stopTAA C. albicans* were cultured on Spider media at 30 and 37 °C for 7 days and assessed for filamentation. (B) Fourfold serial dilutions of wild type, *ume6-EE*, and *ume6-stopTAA C. albicans* were incubated on YPD media at 37 °C for three days. (C) Wild Type and *ume6-stopTAA* incubated in liquid Spider media were assessed for filamentation after 24 hours. (D) $10^5$ cfu of wild type, *ume6-EE*, and *UME6-stopTAA C. albicans* were injected into wax moth larvae, which were subsequently incubated at 37 °C. Survival of larvae was tracked over time. Wt was more virulent than *ume6-EE* ($P = 0.036$) and more virulent than *ume6-stopTAA* as well ($P < 0.001$). Additionally *ume6-EE* was more virulent than *ume6-stopTAA* ($P = 0.003$). Results represent total survival of five independent experiments with 10 larvae per treatment. Survival curves were created using the Kaplan–Meier method and statistical analyses were performed using the log rank test for multiple comparisons (IBM SPSS Statistics; SPSS, Armonk, NY, USA).

conserved C-terminal region are important for *C. albicans* filamentation and virulence. Furthermore, our data suggest residues Lys754 and Lys755 which lie N-terminal to the $Zn^{2+}$ finger domain are important for filamentation and virulence as well.

## DISCUSSION

We have identified restriction enzyme recognition sequences that, when translated, encode consecutive identical amino acids or one or two stop codons. By replacing two codons with these restriction sites, we were able to efficiently introduce and confirm the change of two consecutive amino acids by PCR followed by restriction digestion. In the event that digestion and sequencing are inconclusive, the incorporation of a restriction site also allows facile screening for correct transformants by Southern Blot, a more time consuming, but potentially necessary step to ensure additional copies of a DNA sequence are not present and your mutation of interest has been introduced. We found not all repair templates are incorporated into the genome with equal efficiency. This variability in successful recombination enhances the utility of rapidly screening numerous colonies by digestion. Our technique enabled us to unambiguously, inexpensively, and rapidly identify desired mutations in the *C. albicans* transcription factor *UME6*. Although we did not observe heterozygotes, our restriction strategy should distinguish homozygous from heterozygous mutations. Our data suggest that Ume6 Lys754 and Lys755 are important for filamentation and virulence. In addition, our data suggests the $Zn^{2+}$ finger DNA binding domain and conserved C-terminal domain are also important for filamentation and virulence.

We have identified restriction enzyme recognition sites that encode two consecutive alanines, glutamates, arginines, and glycines and one or two stop codons. Classically, site-directed mutagenesis introduces a single amino acid change, but screening for such changes using restriction digestion is impractical. Introduction of restriction sites with four-base recognition sites (e.g., GCGC which can be cleaved by HinP1I or HhaI) could be used to introduce a single alanine; however, the additional base at the 5′ or 3′ end of the four-base recognition sequence places upstream or downstream sequence constraints that limit its utility.

Introduction of stop codons can be used to truncate the C-terminus of a protein. However, stop codon suppressor mutations could bypass these mutations. The introduction of an I-SceI restriction site introduces two distinct stop codons (TAG and TAA) and is thus less likely to be bypassed with suppressor mutations. We attempted to introduce I-SceI site to *UME6* using a 100-base pair repair template and screened over 50 transformants by restriction digestion. Unfortunately, none of these transformants incorporated the repair template. This apparent inefficiency could be due to slightly shortened regions of homology on the repair template in comparison to the successful repair templates. A moderate increase in repair template size might overcome this limitation. Alternatively, successive SpeI, PacI, or AcuI restriction sites in a single repair template could be used to introduce sequential stop codons and reduce the likelihood of suppression.

All codons are not used equally in genomes. For instance, in *C. albicans* GCN (N representing any base) encodes alanine, but 43% of alanine are encoded by GCT and

less than 5% by GCG (*Arnaud et al., 2005*). Such codon bias must be appreciated when investigators decide which restriction site to introduce. Rare codons may alter protein expression, confounding the interpretation of phenotypes associated with protein variants (*Quax et al., 2015*). We have identified three restriction sites that code for successive alanine residues (Table 2). These enzyme recognition sites are found at different frequencies throughout the genome and allow for the introduction of any alanine codon. This cadre of enzymes provides the investigator flexibility when designing a mutagenesis experiment.

Site-directed mutagenesis restriction digest screening could be exceptionally useful for investigation of protein localization. Proteins use short conserved amino acid sequences to localize to a variety of intracellular compartments including the endoplasmic reticulum, mitochondria, peroxisome, and nucleus (*Nakai, 2000*). These sequence motifs are often charged or hydrophobic. Disruption of such sequence motifs is critical when dissecting the function of these proteins *in vivo*. Site-directed mutagenesis could replace these amino acids with others possessing different biochemical characteristics. Our screening system would enable efficient interrogation of the roles signal motifs play in protein localization.

We piloted our restriction digestion-based strategy to screen for point mutations in *UME6*. Ume6 is a transcriptional regulator of filamentation, and deletion of *UME6* causes defects in filamentation and decreased virulence in mouse models (*Banerjee et al., 2008*). However, little was known about the function of structural elements within Ume6. Ume6 consists of a predicted $Zn^{2+}$ finger DNA binding domain, conserved C-terminal domain, and a poorly conserved N-terminal domain of unknown function (Fig. 1A). Our data show the $Zn^{2+}$ finger domain and/or the C-terminal domain plays a role in filamentation, as introduction of a stop codon upstream of these domains led to decreased filamentation and virulence (Figs. 2A–2C). One potential interpretation is altering conserved lysine residues directly N-terminal to the $Zn^{2+}$ finger domain *causes* defects in filamentation and virulence. This suggests Lys754 and/or Lys755 play a previously unappreciated role in filamentation program activation. Alternatively, changing Lys754 and/or Lys755 could lead to destabilization and/or lower expression of functional protein. The Glu codons that were introduced code for 21% of Glu residues in *Candida* so it is unlikely that codon usage is limiting. While Ume6's N-terminal domain is not as highly conserved as its $Zn^{2+}$ finger or C-terminal domains, it does contain a number of highly conserved residues. Further site-directed mutagenesis of the Ume6 N-terminal domain will help elucidate the role of Ume6 in *C. albicans*' filamentation program.

Site-directed mutagenesis coupled with restriction digest screening has the potential to be broadly useful in a number of ways. First, the development of CRISPR-mediated genome editing systems has significantly expanded the breadth of organisms that can be readily genetically modified (*Reardon, 2016*). These advances are likely to revolutionize knockout library generation in a variety of organisms. For organisms that require a repair template to introduce mutations like *C. albicans* (*Vyas, Barrasa & Fink, 2015*), introduction of stop codons that can be screened by restriction digestion could be an important component of efficient and accurate library preparation and verification. Second, undergraduate research projects are a critical component of education, but identifying projects that both excite students and can be performed during the confines of an undergraduate curriculum is

challenging. This can be especially difficult for students interested in molecular biology, where cloning and reagent development are often times required before one can begin to test a hypothesis experimentally. Introducing mutations using CRISPR and screening for correct mutations using restriction digestion is a quick method whereby undergraduate students can learn and apply multiple molecular biology techniques. The relatively high success rate and moderate cost of these techniques allow students to move projects forward while asking and answering important biological questions. Furthermore, the low cost and speed of these procedures makes these exercises suitable for undergraduate teaching laboratories.

## CONCLUSIONS

We have identified restriction enzymes cut sites that code for successive Alanine, Glutamate, Arginine, Glycine, or stop codons. The incorporation of these sites allows the efficient identification of correct point mutations using restriction digestion. Using CRISPR we introduced these sites to *C. albicans UME6*, changing Lys754 and Lys755 to either two Glutamate residues or a stop codon. We found Lys754 and Lys755 are important for filamentation and virulence. Site-directed mutagenesis coupled with restriction digest screening is a quick cost effective method of screening potential point mutants.

## ACKNOWLEDGEMENTS

The authors would also like to thank Dr. Gennifer Mager and Dr. Eric (VJ) Rubenstein for helpful comments regarding the manuscript.

### Funding

This work was supported by National Institutes Grant 1R15AI130950-01 to Douglas A. Bernstein and an Easter Seal Internship to Olivia L. Smith. The funders had no role in study design, data collection and analysis, decision to publish, or preparation of the manuscript.

### Grant Disclosures

The following grant information was disclosed by the authors:
National Institutes Grant: 1R15AI130950-01.

### Competing Interests

The authors declare there are no competing interests.

### Author Contributions

- Ben A. Evans and Ethan S. Pickerill conceived and designed the experiments, performed the experiments, analyzed the data, contributed reagents/materials/analysis tools, prepared figures and/or tables, authored or reviewed drafts of the paper, approved the final draft.

- Olivia L. Smith conceived and designed the experiments, performed the experiments, analyzed the data, contributed reagents/materials/analysis tools, authored or reviewed drafts of the paper, approved the final draft.
- Mary K. York, Kristen J.P. Buenconsejo and Antonio E. Chambers performed the experiments, contributed reagents/materials/analysis tools, approved the final draft.
- Douglas A. Bernstein conceived and designed the experiments, analyzed the data, contributed reagents/materials/analysis tools, prepared figures and/or tables, authored or reviewed drafts of the paper, approved the final draft.

## Data Availability

The raw data are provided in a Supplemental File.

## Supplemental Information

Supplemental information for this article can be found online at http://dx.doi.org/10.7717/peerj.4920#supplemental-information.

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
