# Peer review of "Restriction digest screening facilitates efficient detection of site-directed mutations introduced by CRISPR in C. albicans UME6"

_PeerJ, doi:10.7717/peerj.4920_

## Round 0.1 · original submission · Minor Revisions

As you see the reviewers are mostly quite positive and give some constructive suggestions. You might want to address some of them in your revised version.

Reviewer 1 ·

Basic reporting

The authors incorporation of restriction enzymes into CRISPR homologous replacement mutagenesis screens is related to previous work using restriction screening for site directed mutagenesis. They should cite Neff et al., Plant J 1998 (which used a similar approach to identify site directed mutations) and Hodgens et al., 2017 (which extended Neff's approach to identify CRISPR indels with their indCAPS tool).

Experimental design

The manuscript reports a clever incorporation of restriction enzymes into a CRISPR mutagenesis screening regime, as well as results of site-directed mutagenesis of the C. albicans UME6 transcription factor. One point of confusion arises from the C. albicans strain used for the study. Was it SC5314 "wildtype"? Also, were the mutations homozygous or heterozygous, as C. albicans is an obligate diploid? Heterozygous mutations would have left 50% cut / 50% uncut on the gel. Were such mutations recovered? Did the authors do anything to remove the pv1093 plasmid after CRISPR mutagenesis? If not, heterozygotes would repair randomly to homozygous wt or homozygous mutant, right?

Validity of the findings

While the experiments were well done and appropriate controls and statistical tests were performed, the authors overstate the results in the discussion line 223. Their results don't show that UME6 is "essential for filamentation", as they don't show any defect in hyphal growth in figure 2. There may be such a defect (visible under the microscope, e.g.). If so, please show it. In addition, there wasn't a "dramatic loss of filamentation and virulence". Virulence was reduced, not lost. This is just overstated. Probably "led to decreased filamentation and virulence" would be fine, assuming they show some change in filamentation beyond altered colony morphology.

Additional comments

Generally nice paper. Adding references to previous restriction enzyme screening approaches and lightening the interpretation of UME6 mutants a bit would make this stronger. Also, the authors should revise the manuscript to clarify the distinction between homozygous and heterozygous mutants.

Reviewer 2 ·

Basic reporting

No comment.

Experimental design

No comment.

Validity of the findings

No comment.

Additional comments

The work "Restriction digest screening facilitates efficient detection of site-directed mutations by CRISPR in Candida albicans" by Evans and colleagues describes a useful mutagenesis and screening strategy for the important fungal pathogen Candida albicans. While mutagenesis at some loci is very efficient, others are more difficult, requiring extensive screening to identify successful mutants. The approach presented can significantly reduce the cost of identification of such mutants. Though not emphasized, the authors have additionally identified mutants which impact filamentation and pathogenesis.

A few minor suggestions for the authors, in no particular order:
1. The authors discuss the importance of codon usage in designing a species specific mutagenesis strategy (in paragraph starting at line 200), but do not discuss the codon usage for the codons they chose. Perhaps worth noting if the codon frequency was/was not limiting for the E codons they chose.

2. C. albicans is a diploid, therefore successful CRISPR mutagenesis requires mutation of both copies of a gene - even more for guides targeting multiple genes. A key advantage of the restriction strategy proposed is that it can sensitively/reliably detect mixtures, which would result in incomplete digestion. Such mixtures are not always reliably detected by sequencing, where mixed base calls rely on more ideal templates. This additional benefit of the authors' strategy might be worth highlighting in the discussion.

3. The rationale for mutagenesis of the N-terminal domain of UME6 at codons 754 and 755 was not given. Is this a conserved region of the protein? Is there a predicted function? Have other studies been done to suggest this is important, or is there any clear reason why the mutants chosen would have the hypomorphic phenotype shown?

4. As this article is intended to provide some guidance for mutagenesis strategies, perhaps it would be worthwhile to depict the guide in Figure 1C-E, and how mutagenesis destroys the recognition motif. If so may be worth showing Figure 1E as "WT" sequence, rather than "failed mutant," and simply note the failed mutant had WT sequence. If so, it should be shown before mutant sequences.

There were a few minor errors in the text noted below:
Line 67: "reviewed in" should be inside the reference parenthesis
Line 117: C. albicans instead of Candida albicans
Line 146: The statement refers to data shown at 2A, not 2B.
Table 2 (last line, last column): should be I-SceI not 1-SceI

·

Basic reporting

Figure are not relevant in light of the title. No figure shows any restriction digest, and one figure relates to the function of a gene, again not the topic of the title. I would suggest changing the title.

Otherwise, the basic reporting is fine.

Experimental design

No Comment.

Validity of the findings

Any conclusion about the importance of KK to the function of ume6 is premature without verification of expression by something like western blot. If the KKíEE mutation causes a loss of soluble expression, no conclusion about the functionality of amino acids is possible.

Introduction of EE mutation appears to introduce multiple PAM sites. Sequencing of many clones is needed to analyze the non-RE digested samples, that might have incorporated the mutations but later mutated the site again by Cas9 activity.

Additional comments

I don't really understand what type of scanning mutatgenesis library is proposed? I am not familiar with applications where two adjacent alanines are needed. A reference would help.

Can you add the CRISPR target site to your Figure 1A?

How does this work to make scanning libraries, as suggested?

---

## Round 0.2 · accepted · Accept

I think you have addressed all concerns by the reviewers appropriately. Thanks for the quick revision.

#